# Continuous NPWT Regulates Fibrosis in Murine Diabetic Wound Healing

**DOI:** 10.3390/pharmaceutics14102125

**Published:** 2022-10-06

**Authors:** Mengfan Wu, Dany Y. Matar, Zhen Yu, Ziyu Chen, Samuel Knoedler, Brian Ng, Oliver A. Darwish, Sadaf Sohrabi, Leigh Friedman, Valentin Haug, George F. Murphy, Yuval Rinkevich, Dennis P. Orgill, Adriana C. Panayi

**Affiliations:** 1Department of Plastic Surgery, Peking University Shenzhen Hospital, Shenzhen 518036, China; 2Division of Plastic Surgery, Department of Surgery, Brigham and Women’s Hospital, Harvard Medical School, Boston, MA 02115, USA; 3School of Medicine, Washington University in St. Louis, St. Louis, MO 63110, USA; 4Shenzhen Key Ophthalmic Laboratory, Department of Ophthalmology, Shenzhen Eye Hospital, Jinan University, Shenzhen 518040, China; 5Angiogenesis Laboratory, Department of Ophthalmology, Massachusetts Eye and Ear Infirmary, Harvard Medical School, Boston, MA 02114, USA; 6Department for Plastic Surgery and Hand Surgery, Klinikum Rechts der Isar, Technical University of Munich, 81675 Munich, Germany; 7School of Medicine, Saint Louis University, St. Louis, MO 63104, USA; 8School of Medicine, California Northstate University, Elk Grove, CA 95757, USA; 9Division of Dermatopathology, Department of Pathology, Brigham and Women’s Hospital, Harvard Medical School, Boston, MA 02115, USA; 10Department of Medicine, Lehigh University, Bethlehem, PA 18015, USA; 11School of Medicine, Drexel University, Philadelphia, PA 19129, USA; 12Department of Hand, Plastic and Reconstructive Surgery, Microsurgery, Burn Center, BG Trauma Center Ludwigshafen, University of Heidelberg, 67071 Ludwigshafen, Germany; 13School of Medicine, Helmholtz Zentrum München, Institute of Regenerative Biology and Medicine, 81377 Munich, Germany

**Keywords:** wound healing, fibrosis, scarring, NPWT, mechanotransduction, YAP, tissue regeneration, caspase3

## Abstract

Scarring is associated with significant morbidity. The mechanical signaling factor yes-associated protein (YAP) has been linked to Engrailed-1 (En1)-lineage positive fibroblasts (EPFs), a pro-scarring fibroblast lineage, establishing a connection between mechanotransduction and fibrosis. In this study, we investigate the impact of micromechanical forces exerted through negative pressure wound therapy (NPWT) on the pathophysiology of fibrosis. Full-thickness excisional dorsal skin wounds were created on diabetic (db/db) mice which were treated with occlusive covering (control) or NPWT (continuous, −125 mmHg, 7 days; NPWT). Analysis was performed on tissue harvested 10 days after wounding. NPWT was associated with increased YAP (*p* = 0.04) but decreased En1 (*p* = 0.0001) and CD26 (*p* < 0.0001). The pro-fibrotic factors Vimentin (*p* = 0.04), α-SMA (*p* = 0.04) and HSP47 (*p* = 0.0008) were decreased with NPWT. Fibronectin was higher (*p* = 0.01) and collagen deposition lower in the NPWT group (*p* = 0.02). NPWT increased cellular proliferation (*p* = 0.002) and decreased apoptosis (*p* = 0.03). Western blotting demonstrated increased YAP (*p* = 0.02) and RhoA (*p* = 0.03) and decreased Caspase-3 (*p* = 0.03) with NPWT. NPWT uncouples YAP from EPF activation, through downregulation of Caspace-3, a pro-apoptotic factor linked to keloid formation. Mechanotransduction decreases multiple pro-fibrotic factors. Through this multifactorial process, NPWT significantly decreases fibrosis and offers promising potential as a mode to improve scar appearance.

## 1. Introduction

Deep and voluminous skin injuries in mammals and humans commonly leads to a fibrotic response and ultimately to scar tissue formation [1]. Extensive scarring, such as that seen in hypertrophic and keloid scarring, is a major medical problem and a debilitating sequela of surgery [2]. These pathologies have significant functional and esthetic morbidity and severely impact patients’ quality of life. This is in contrast to lower vertebrates, superficial wounds, or early gestation fetal skin wounds which heal by regeneration and re-establishment of the normal dermal structure [3,4,5,6]. The ability to shift the response from scarring to pro-regenerative in adult skin would undoubtedly revolutionize medicine.

Prior research has identified that the fibrotic process that occurs in response to skin wounding is mediated by a specific lineage of fibroblasts termed Engrailed-1 (En1)–lineage positive fibroblasts (EPFs), where Engrailed-1 (En1) is a transcription factor that is exclusively expressed in scar-prone fibroblasts that are responsible for scar deposition [7]. In addition to EPFs, embryonic and adult skin contains En1-lineage negative fibroblasts (ENFs), a functionally distinct lineage of fibroblasts that does not express En1 nor contributes to adult scar production, but which regenerates the healthy lattice organization during fetal wound healing [8,9]. During postnatal life and during adult wound healing, the abundance of EPFs increases over ENFs, which contributes to the scar phenotype seen in adult wounds [10,11].

The process underlying the abundance and activation of EPFs was indicated to depend on mechanical cues and mechanotransduction. Mascharak et al. cultured ENFs on plastics of varying stiffness to investigate how mechanical cues affect En1 expression in ENFs [11]. Reticular ENFs cultured on soft hydrogels did not indicate activation of En1, whereas those cultured on high-stiffness plastic indicated upregulated En1 expression and transitioned to EPFs. Inhibition of Rho-kinase, a mediator of mechanotransduction, inhibited EPF activation [11,12]. Furthermore, inhibition of the mechanical signaling factor yes-associated protein (YAP), either genetically or pharmacologically, with the use of verteporfin, was indicated to sufficiently inhibit EPF numbers and activation states and, consequently, decrease scarring in murine wound healing [11].

A caveat of the aforementioned research is that Negative Pressure Wound Therapy (NPWT), which is a well-established therapy in the treatment of traumatic, surgical, and chronic wounds, depends on mechanotransduction and has been demonstrated clinically to result in decreased scarring, particularly when used in incisional wounding [13,14,15,16,17,18,19,20,21,22,23]. Pre-clinical studies on porcine surgical wounds also reported an overall improved scar appearance with the use of NPWT [24,25,26,27]. Despite the multiple promising clinical and pre-clinical studies, these studies focused on final scar appearance without investigating the effect on scarring pathophysiology. An investigation into the molecular mechanisms that underlie the actions of NPWT on scarring and fibrosis is lacking from the literature.

Furthermore, most studies investigated the effect of NPWT in incisional wounding. Incisional wounds occur during surgery when skin, muscle and subcutaneous tissue are incised whereas excisional wounds occur when a section of full-thickness skin is entirely removed from the wound bed. Differences exist between incisional and excisional wounds, particularly with regards to their ability to respond tension [28]. Specifically, application of minimal tension to incisional wounds results in stretching of elastin and buckling of collagen, revealing mostly elastin, whereas application of the same tension to excisional wounds, results in the reverse, that is, stretching of collagen and buckling of elastin, revealing mainly collagen [28,29]. To the best of our knowledge, there are currently no published studies investigating the effects of NPWT on the fibrosis pathwat in excisional wounds. Therefore, current evidence on NPWT’s effect on the fibrotic pathway at the molecular level, in addition to on excisional wounding, is lacking.

We hypothesize that modulation of the wound microenvironment through mechanotransduction forces, such as those applied by NPWT, could be a valid method to minimize pathological scarring. We therefore used a murine diabetic excisional model to investigate the effects of NPWT on multiple factors in the pathway of scarring pathology, including YAP, En1, adenosine deaminase complexing protein 2 CD26 and smooth muscle alpha-actin (α-SMA). We investigate diabetic mice because they are known to display poor wound healing, and chronic refractory wounds have been characterized to result in hypertrophic scars [30,31].

## 2. Materials and Methods

### 2.1. Animals

Homozygous, genetically diabetic (db/db) mice (Jackson Laboratories; Bar Harbor, ME, USA) were purchased at 10 weeks and acclimatized for one week in the animal facility of Brigham and Women’s Hospital prior to surgery. Animals were single-housed with cage enrichment, and food and water were provided ad libitum. All animal procedures were performed according to a protocol approved by the Institutional Animal Use and Care Committee of Harvard Medical School (Number: 2018N000117).

### 2.2. Surgery and Post-Surgical Monitoring

Twenty db/db mice underwent hair removal (Nair^®^; Church & Dwight Co., Ewing Township, NJ, USA) in preparation for surgery. Vapor anesthesia was induced with 2–4% isoflurane in an induction chamber. A single full-thickness 1 × 1 cm^2^ dorsal wound, with excision of the panniculus carnosus was created on the dorsum of each mouse. Ten mice (control group; Figure 1) received an occlusive dressing wound coverage. Ten mice were treated with a NPWT setup set at a pressure of −125 mmHg, applied continuously for 24 h/day for seven days (NPWT group). Briefly, to create the NPWT setup, a 1.5 cm^2^ square window was created in a 2.5 cm^2^ DuoDERM^®^ Dressing (ConvaTec, Reading, UK), and the adhesive side of the DuoDERM^®^ was applied directly to the skin securing the wound in the center of the window. A 1 cm^3^ cube of polyurethane foam (GranuFoam; Kinetic Concepts Inc., San Antonio, TX, USA) was applied over the wound and adhesive film (V.A.C.^®^ Therapy, Kinetic Concepts Inc.) was used to seal the foam ensuring to cover the entire setup. A tube inserted in the foam was connected to a vacuum source (V.A.C.^®^ Therapy System, Kinetic Concepts Inc.). A final layer of occlusive dressing (Tegaderm™; 3M™, St. Paul, MN, USA) was placed over the wound setup and fixed in place with corner sutures (4–0 Prolene suture, Ethicon, Inc., Somerville, NJ, USA) on the dorsum (Figure 1). A strict check-up schedule was followed to ensure vacuum was continuously maintained, with assessments every two hours. If the device was not sealed properly, an alarm would sound, and the dressings would be modified accordingly. On day 3 post-surgery, the entire apparatus, including the foam and dressings, was changed and fresh materials were applied as described above. After seven days of suction, the NPWT setup was replaced with simple occlusive dressing. The wound and 5 mm in the wound perimeter were harvested upon sacrifice. The tissue was split in half and was either fixed for histology using a formalin-ethanol protocol or cryopreserved for biochemical analysis using liquid nitrogen.

### 2.3. Histology

All histological analysis was performed using five-micron-thick cross-sections of paraffin- embedded tissue of samples taken on day 10 post-surgery. Masson’s Trichrome (MT) staining was performed according to standard protocol [32]. For immunofluorescent staining, de-paraffinized and rehydrated cross-sections were probed with the antibodies against YAP (1:200; PA1-46189, Invitrogen, Waltham, MA, USA), PDZ-binding motif (TAZ 1:300, NB110-58359SS, Novus Biologicals, Centennial, CO, USA), En1 (1:200, BS-11744R, Bioss Antibodies, Woburn, MA, USA), α-SMA (1:100; NBP1-30894, Novus Biologicals, Centennial, CO, USA), CD26 (1:150, 559652, BD Biosciences, Haryana, India), Fibronectin (1:400, NBP1-91258SS, Novus Biologicals, Centennial, CO, USA), Ki67 (1 ug/mL, ab15580, Abcam, Cambridge, UK), Heat shock protein 47 (Hsp47, 1:200, ab109117, Abcam, Cambridge, UK), Vimentin (1:200, ab92547, Abcam, Cambridge, UK) or S100A4 (1:200, ab197896, Abcam, Cambridge, UK) for 16 h at 4 °C. Bovine serum albumin (BSA) (5%; one hour) was used as the blocking agent and the slides were incubated overnight at 4 °C with the primary antibody. The following day the slides were thoroughly washed and incubated for one hour in the dark with the secondary goat anti-rabbit antibodies (1:400; Thermo Fisher, Waltham, MA, USA). To assess apoptosis, the sections were stained using a terminal deoxynucleotidyl transferase (TdT)–mediated dUTP nickend labeling (TUNEL) assay following the manufacturer’s instructions (Thermo Fisher Scientific, Waltham, Waltham, MA, USA). Last, nuclei were stained and the sections mounted with ProLong^®^ Diamond Antifade Mountant with DAPI (P36971, Invitrogen). For all assessments, three high power field images (HPF) of the wound (border or bed) of each cross-section were taken per sample (*n* = 10 per group) with an Olympus microscope (BX53 UCMAD3; T7, Tokyo, Japan). MT stained section were used to quantify collagen deposition by applying the image thresholding function in ImageJ software (version 1.52a; Media Cybernetics, Rockville, MD, USA), as previously described [33]. Fibronectin, Hsp47, Vimentin and S100A4 were expressed as relative fluorescence intensity. TAZ, En1 or Ki-67–positive cells were counted and expressed as a ratio of positive cells to total cells (stained with DAPI).

### 2.4. Western Blotting

A Tissue Protein Extraction Reagent (78510; Thermo Scientific, Waltham, MA, USA) containing protease inhibitors (11836170001; MilliporeSigma; Burlington, MA, USA) was used to extract protein from the cryopreserved tissue. Concentration was quantified with Coomassie Plus (Bradford) Assay Reagent (23238; Thermo Scientific, Waltham, MA, USA). The wound tissue was homogenized through multiple centrifugations and aspirations of the supernatant. Samples were heated (90 °C, six minutes), loaded onto Bis-Tris NuPAGE gels (NP0321PK2; Thermo Scientific, Waltham, MA, USA) and the proteins in the gels were transferred to a polyvinylidene difluoride (PVDF) membrane (IPVH304F0; MilliporeSigma; Burlington, MA, USA) which was blocked with 5% BSA (0.1% Tween 20, room temperature, 1.5 h). The membranes were incubated at 4 °C overnight with anti-RhoA (1:1000, 2117, CST, MA, USA), anti-YAP1 (1:1000, PA1-46189, Thermo Scientific, Waltham, MA, USA) or anti-Caspase-3 (1:1000, 9662, CST, MA, USA). This was followed by a 1 h incubation at room temperature with horseradish peroxidase (HRP)-conjugated secondary antibodies. Finally, after washing (0.1% Tween 20) enhanced chemiluminescence (ECL) was used to detect antibody bindings and densitometry was performed using ImageJ (version 1.52a; Media Cybernetics, Rockville, MD, USA). β-Actin (1:1000, 4970S, Cell Signaling Technology. Danvers, MA, USA) and Glyceraldehyde-3-phosphate dehydrogenase (GAPDH) (1:1000, 5174, Cell Signaling Technology. Danvers, MA, USA) served as the internal reference controls.

### 2.5. Statistical Analysis

Data are expressed as means ± standard deviation. Statistically significant differences between the two groups were estimated with a Student’s *t*-test with significance set at a *p*-value < 0.05. All analysis and graphical presentation were performed in GraphPad Prism (Version 8.00; MacOS, GraphPad Software, La Jolla, CA, USA).

## 3. Results

### 3.1. NPWT Was Associated with an Increase in YAP but a Decrease in Pro-Fibrotic Fibroblasts

The relative fluorescence intensity (RFI) of YAP was significantly higher in the NPWT group (2.3 ± 1.3 RFI) compared to the control (1 ± 0.6 RFI; *p* = 0.04; Figure 2a,b). No differences were found between the relative fluorescence intensity of TAZ in the NPWT group (16.9 ± 11.8 RFI) and the control (20 ± 9.9 RFI; *p* = 0.58; Figure 2a,c). The percentage of En1 positive cells in the NPWT group (2.8 ± 2.6% En1 + cells) was significantly lower than the control (37 ± 32% En1 + cells; *p* = 0.01; Figure 2a,d). In murine skin, most scar formation is caused by CD26-positive fibroblasts. The percentage of CD26-positive cells in the NPWT group (13 ± 5% CD26 + cells) was significantly lower than the control (25 ± 3% CD26 + cells; *p* < 0.0001; Figure 2a,e).

### 3.2. NPWT Modified Collagen Deposition

Collagen deposition was significantly lower in the NPWT group (24 ± 7.7%) compared to the control (33 ± 7.1; *p* = 0.02; Figure 3a,b). Collagen density was similar between the two groups (NPWT: 6.1 ± 3.6 × 10^7^ vs. control: 5.9 ± 2.7 × 10^7^; *p* = 0.86; Figure 3a,c). Likewise, collagen cell count did not differ between the two groups (NPWT: 390 ± 90 cells/HPF vs. control: 480 ± 200 × 10^7^ cells/HPF; *p* = 0.26; Figure 3a,d).

### 3.3. NPWT Was Associated with Increased Fibronectin, but Decreased αSMA, Vimentin and Hsp47

Fibronectin was found to be significantly higher in the NPWT group (2.2 ± 1 RFI) compared to the control (1 ± 0.6; *p* = 0.01; Figure 4a,b). Hsp47 was significantly lower in the NPWT (12 ± 5.6%) compared to the control group (31 ± 9.2%; *p* = 0.0008; Figure 4a,c). Vimentin was significantly lower in the NPWT (160 ± 50 RFI) compared to the control group (330 ± 200 RFI; *p* = 0.04; Figure 4a,d). αSMA deposition followed a similar pattern, and was significantly lower in the NPWT (16 ± 5% αSMA + cells) compared to the control group (24 ± 8%αSMA + cells; *p* = 0.04; Figure 4a,b). There were no significant differences in the amount of S100A4 in the NPWT (130 ± 80 RFI) compared to the control group (140 ± 50 RFI; *p* = 0.73; Figure 4a,e).

### 3.4. NPWT Affected Cellular Turnover

Cellular proliferation of the wound bed was significantly higher in the NPWT group (14 ± 7.2% Ki67 + cells/HPF) compared to the control group (5.2 ± 2.4% Ki67 + cells; *p* = 0.002; Figure 5a,b).

Cellular proliferation of the wound border was also significantly higher in the NPWT group (19 ± 14% Ki67 + cells/HPF) compared to the control group (7.3 ± 3.5% Ki67 + cells; *p* = 0.03; Figure 5a,c). Apoptosis was significantly different between the groups, with the control group having more apoptotic cells (29 ± 10% TUNEL +/HPF) than the NPWT group (19 ± 6% TUNEL +/HPF; *p* = 0.03; Figure 5a,d).

### 3.5. NPWT Upregulated RhoA and YAP, while Downregulating Caspase-3

Western blotting identified statistically significant differences in the ratio of YAP to GAPDH, with the NPWT group having a significantly higher ratio (1.8 ± 1) than the control group (1 ± 0.5; *p* = 0.02) on day 10 (Figure 6a,b). The ratio of RhoA to GAPDH was also significantly higher in the NPWT group (1.8 ± 0.4) compared to the control group (1 ± 0.5; *p* = 0.03) on day 10 (Figure 6a,c). On the other hand, the ratio of Caspase-3 to β-Actin was significantly lower in the NPWT (0.9 ± 0.4) than the control group (1.7 ± 0.7; *p* = 0.03) on day 10. At the protein level, this suggests that NPWT upregulated the expression of RhoA and YAP, whereas Caspase-3 was downregulated (Figure 6a,d).

## 4. Discussion

In this study we applied NPWT to a murine excisional wound model to investigate its effect on the pathophysiology of fibrosis and identified its impact on multiple components of the fibrotic pathway. Specifically, we saw that NPWT led to an increase in YAP, fibronectin, RhoA, and cellular proliferation, and a decrease in En1, CD26, collagen, vimentin, Hsp47, caspase-3 and cellular apoptosis in murine excisional wounds.

Our results regarding decreased caspase-3 and apoptosis following NPWT treatment agree with prior research on human keloid and hypertrophic scarring. The involvement of caspase-3 and apoptosis in pathological scarring was previously studied by Akasaka et al., who quantified the expression of caspase-3 in surgically resected scar tissues of three different durations, <10 months, 10 to 40 months, >40 months old [34]. Compared to normally healed flat scars, increased levels of caspase-3 were seen in the hypertrophic scars and keloids of all three groups. In addition, the team found, and verified in a later study, a higher number of TUNEL-positive cells in hypertrophic scars and keloid, particularly in the hypertrophic scars and keloid that were older than 10 months [34,35]. Thus, our results suggest that NPWT’s ability to decrease caspase-3 and apoptosis contributes to improved scarring outcomes in excisional wounds.

Additionally, we identified decreased vimentin and a difference in its overall spatial distribution following NPWT treatment. Vimentin has been previously linked to scarring phenotypes in corneal injury [36], and in bleomycin-induced lung fibrosis [37]. Also, whereas in the NPWT group, vimentin appeared to be staining individual cells (i.e., fibroblasts), in the control group it appeared to bind in a less cellular but more extracellular, filamentous manner. Although there are many gaps in our understanding of the role of vimentin in fibrosis, prior research has indicated that extracellular vimentin can specifically bind to mesenchymal leader cells to signal their phenotypic change to myofibroblasts in a profibrotic environment [38]. Following injury, the differentiation of mesenchymal cells to myofibroblasts, is believed to underly fibrosis and scarring [39]. In addition, hypertrophic scarring followed by fibrosis induced by TNF-α has been demonstrated to result in an increase in S100A4 and vimentin expression [40]. Although our results indicate that NPWT results in a decrease in vimentin and more cellular localization of vimentin, which agrees with prior studies demonstrating such changes in decreased scarring phenotypes, we saw no differences in S100A4, highlighting the need for further studies in the pathophysiology of fibrosis and the involvement of the different factors. Furthermore, CD26 has previously been established as a marker of fibroblasts with increased fibrotic activity [7]. In this study, NPWT was associated with a decrease in the presence of CD26-positive fibroblasts, again agreeing with prior research on CD26′s role during fibrosis.

Furthermore, our study demonstrated decreased Heat shock protein 47 (HSP47) expression following NPWT treatment, which is a collagen-specific chaperone required for the maturation of pro-collagen to collagen and has been previously proven to be involved in excessive collagen accumulation seen in scar tissue [41]. Treatment of cultured fibroblasts with exogenous Hsp47 has been demonstrated to promote collagen deposition [42,43]. In addition, Wang et al. induced dorsal wounds in fetal rat skin and concluded that the scarless healing of fetal skin is related to a lack of change in Hsp47 expression [44]. Interestingly, pro-collagen is the only substrate for Hsp47, highlighting the potential promise of using NPWT as an anti-fibrotic therapy [45].

Taken together, our results indicate that NPWT interferes with the normal fibrotic response (Figure 7). In non-fibrotic conditions, cytoplasmic YAP bound to a-catenin via 14-3-3 is not able to translocate into the nucleus [46]. In a pro-fibrotic environment, caspase-3 is upregulated and cleaves a-catenin [46], allowing an increase in YAP expression and its translocate to the nucleus where it promotes En1 transcription, inducing a switch from pro-regenerative ENFs to pro-fibrotic EPFs [11]. A pro-fibrotic response promotes the conversion of pro-collagen to collagen and the increased collagen deposition further amplifies the loop, resulting in excessive scarring [47]. In this study, we demonstrate for the first time the uncoupling of YAP and En1; although NPWT increased the expression of YAP, the expression of En1 was decreased. In addition, although factors found on the fibrotic cycle pre-YAP sequestration, such as fibronectin and RhoA were increased, factors found on the cycle post-YAP sequestration, such as En1, Hsp47, and collagen deposition were decreased. This identifies that the plausible underlying mechanism behind NPWT’s uncoupling effect is a lack of YAP nuclear sequestration, which may be explained by NPWT’s downregulation of Caspase-3 which we identified. Decreased caspase-3 would result in a decrease in the cleavage of a-catenin, a decrease in nuclear sequestration of YAP and finally a decrease in En1 transcription.

One limitation of this study was the use of murine models. Although wound healing physiology is not homologous across rodents and humans, steps can be taken to mimic human wound healing physiology in murine models. The largest limitation facing murine wound healing models is the tendency of murine wounds to close mainly by wound contraction, while normal human wound closure mainly occurs via epithelization. In our study we attempted to avoid contraction-mediated wound healing by removing the panniculus carnosus during wound excision, the thin sheet of striated muscle that lies between the subcutaneous fat and dermal layers that is responsible for skin contraction. As a result, histological analysis demonstrated reduced wound contraction by the significant presence of granulation tissue and migratory epidermal cells in both treatment groups, which signifies a wound healing process like human skin and allows us to make general conclusions about normal human wound healing. Similarly, we chose to quantitatively assess the wound healing by using tissue harvested 10 days post-surgery, this point has been previously identified to be at a stage pre-epithelial closure. Specifically, Chen et al. demonstrated that although murine wounds heal both by contraction and re-epithelialization, most contraction occurs post epithelial closure [48]. At the same time, although there are many novel reconstructed human skin models on the market, such as the ex-vivo model NativeSkin which appears to be the closest model mimicking normal human skin physiology, such models are limited by their smaller surface areas and shorter lifespans. Another limitation of this study was the limited investigation of scarring pathway markers. In this study we investigated in detail the YAP-En1 mechanotransduction pathway, however, there still exists a variety of other metabolic and physiological mechanisms that were not investigated and that may contribute to NPWT’s ability to modulate scarring. It is important to investigate other complimentary pathways in the future to supplement our results.

Vimentin has been previously linked to scarring phenotypes in corneal injury [36], and in bleomycin-induced lung fibrosis [37]. In this study, we identified an increase in the expression of vimentin in the control group and a difference in its overall distribution. Whereas in the NPWT group, vimentin appeared to be staining individual cells (i.e., fibroblasts), in the control group it appeared to bind in a less cellular but more extracellular, filamentous manner. Although there are many gaps in our understanding of the role of vimentin in fibrosis, prior research has indicated that extracellular vimentin can specifically bind to mesenchymal leader cells to signal their phenotypic change to myofibroblasts in a profibrotic environment [38]. Following injury, the differentiation of mesenchymal cells to myofibroblasts, is believed to underly fibrosis and scarring [39]. In addition, hypertrophic scarring followed by fibrosis induced by TNF-α has been proven to result in an increase in S100A4 and vimentin expression [40]. We demonstrate that NPWT results in a decrease in vimentin but not in S100A4, highlighting the need for further studies in the pathophysiology of fibrosis and the involvement of the different factors. Furthermore, CD26 has previously been established as a marker of fibroblasts with increased fibrotic activity [7]. In this study, NPWT was associated with a decrease in the presence of CD26-positive fibroblasts.

Furthermore, increased expression of Heat shock protein 47 (HSP47), a collagen-specific chaperone required for the maturation of pro-collagen to collagen, has been previously proven to be involved in excessive collagen accumulation seen in scar tissue [41]. Treatment of cultured fibroblasts with exogenous Hsp47 has been proven to promote collagen deposition [42,43]. In addition, Wang et al. induced dorsal wounds in fetal rat skin and concluded that the scarless healing of fetal skin is related to a lack of change in Hsp47 expression [44]. Interestingly, pro-collagen is the only substrate for Hsp47, highlighting the potential promise of using NPWT as an anti-fibrotic therapy [45].

Taken together, our results indicate that NPWT interferes with the normal fibrotic response (Figure 7). In non-fibrotic conditions, cytoplasmic YAP bound to a-catenin via 14-3-3 is not able to translocate into the nucleus [46]. In a pro-fibrotic environment, caspase-3 is upregulated and cleaves a-catenin, [46] allowing YAP to translocate to the nucleus where it promotes En1 transcription, inducing a switch from pro-regenerative ENFs to pro-fibrotic EPFs [11]. A pro-fibrotic response promotes the conversion of pro-collagen to collagen and the increased collagen deposition further amplifies the loop, resulting in excessive scarring [47]. In this study, we demonstrate that although NPWT increased the expression of YAP, the expression of En1 was decreased. In addition, although factors found on the fibrotic cycle pre-YAP sequestration, such as fibronectin and RhoA were increased, factors found on the cycle post-YAP sequestration, such as En1, Hsp47, and collagen deposition were decreased. This identifies that the plausible underlying mechanism is a lack of YAP nuclear sequestration, which may be explained by the decrease in Caspase-3 which we identified. Decreased caspase-3 would result in a decrease in the cleavage of a-catenin, a decrease in nuclear sequestration of YAP and finally a decrease in En1 transcription.

### Innovation

Although the current accepted hypothesis is that mechanotransduction, which involves increased tension between cell and the extracellular matrix, leads to upregulation of YAP and increased En1 transcription [11,47], we demonstrate in this study that NPWT uniquely decouples YAP from En1.

Furthermore, our findings indicate that the mechanotransduction forces exerted by NPWT decrease multiple pro-fibrotic factors, including En1, collagen, vimentin, Hsp47, and caspase-3. Through these actions, we identify how NPWT may play a significant role in improving final scar appearance. We hope this study will spur more research investigating the pathways underlying NPWT’s anti-scarring effects and encourage clinicians to utilize NPWT after excisional-type wounds to modulate fibrosis and scarring after injury. In the future, we hope to investigate the effect of NPWT on incisional wounds and burns, and also understand the physiological and phenotypic outcomes of combining NPWT with other fibrosis-decreasing agents such as verteporfin.

## 5. Conclusions

In this study we used a murine experimental model, to elucidate for the first time the mechanisms of action of NPWT on the signaling pathways involved in fibrosis. We identify that NPWT downregulates Caspace-3, a pro-apoptotic factor linked to keloid formation. In addition, NPWT results in an increase in YAP but, paradoxically, a decrease in EPFs, appearing to uncouple YAP from pro-fibrotic fibroblasts. Overall, the mechanotransduction forces exerted by NPWT decrease multiple pro-fibrotic factors. As such, NPWT holds promising potential as a scarring modulator.

## Figures and Tables

**Figure 1 pharmaceutics-14-02125-f001:**
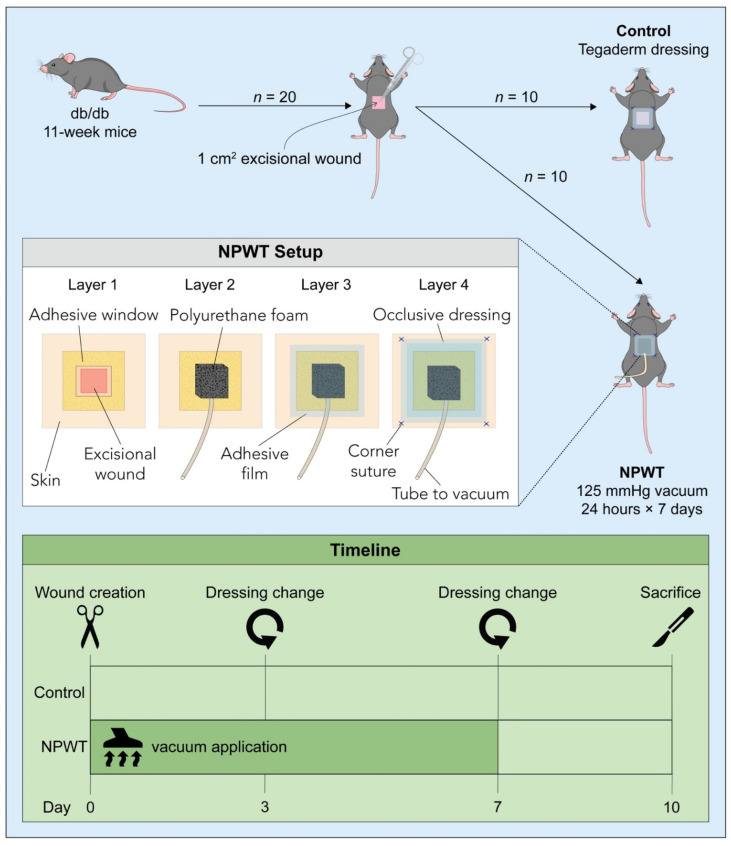
Study design. 20 db/db mice received a full-thickness dorsal skin excisional wound (1 × 1 cm). 10 mice were treated with NPWT (NPWT group), and 10 mice were covered with an occlusive dressing (control group). The NPWT setup displays the setup of the different layers and the timeline depicts the time points of NPWT application, dressing change and sacrifice.

**Figure 2 pharmaceutics-14-02125-f002:**
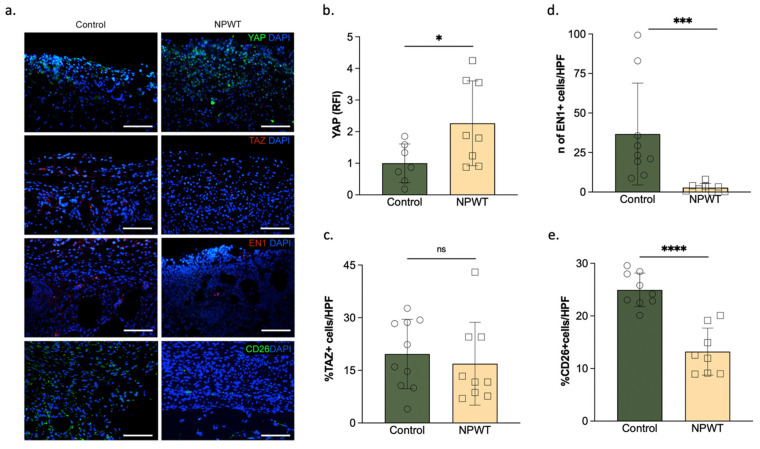
YAP, TAZ, En1 and CD26 levels. (**a**) Immunofluorescent staining of the wound bed. anti-YAP, anti-TAZ, anti-En1 and anti-CD26 with DAPI stained sections of the wound bed of the two groups on day 10. Scale bar = 100 µm. (**b**) Relative fluorescence intensity of YAP. YAP presence significantly differed between the two groups (NPWT: 2.3 ± 1.3 RFI vs. control:1 ± 0.6; *p* = 0.04). (**c**) Percentage of TAZ + ve cells. There were no differences in the presence of TAZ between the two groups. (**d**) Percentage of En1 + ve cells. En1 was significantly lower in the NPWT group (NPWT:2.8 ± 2.6% En1 + cells vs. control:37 ± 32% En1 + cells; *p* = 0.01). (**e**) Percentage of CD26 + ve cells. CD26 was significantly lower in the NPWT group (NPWT: 2.8 ± 2.6% CD26 + cells vs. control:37 ± 32% CD26 + cells; *p* < 0.0001). Number of circles/squares per bar equals sample size (*n* = 7–10) with each circle/square representing the average of three HPF measurements. Where ns equals *p* > 0.05, * equals *p* ≤ 0.05, *** equals *p* ≤ 0.001 and **** equals *p* ≤ 0.0001.

**Figure 3 pharmaceutics-14-02125-f003:**
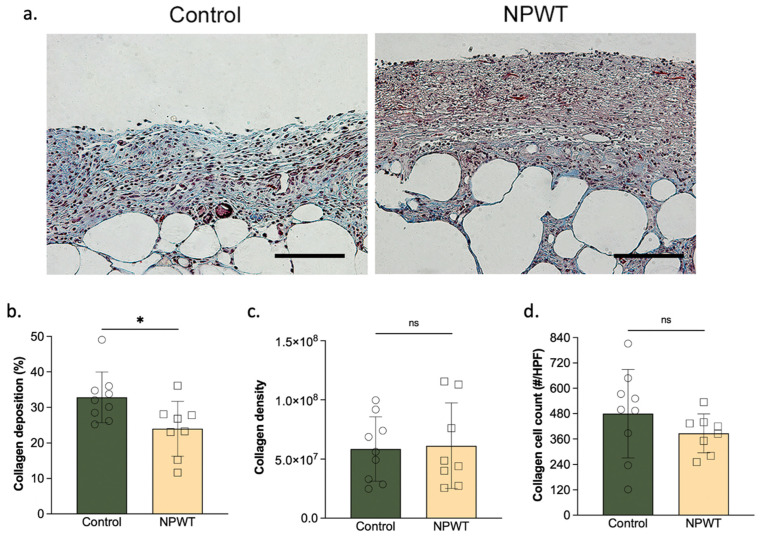
Collagen deposition. (**a**) Representative sections. MT-stained wound beds of the two groups on day 10. Scale bar = 100 µm. (**b**) Collagen deposition. Collagen deposition significantly differed between the two groups (NPWT: 24 ± 7.7% vs. control: 33 ± 7.1; *p* = 0.02). (**c**) Collagen density. There were no significant differences in collagen density between the two groups (NPWT: 6.1 ± 3.6 × 10^7^ vs. control: 5.9 ± 2.7 × 10^7^; *p* = 0.86). (**d**) Collagen cell count. There were no significant differences in collagen cell count (NPWT: 387 ± 91 cells/HPF vs. control: 480 ± 208 × 10^7^ cells/HPF; *p* = 0.26). Number of circles/squares per bar equals sample size (*n* = 8–9) with each circle/square representing the average of three HPF measurements. Where ns equals *p* > 0.05 and * equals *p* ≤ 0.05.

**Figure 4 pharmaceutics-14-02125-f004:**
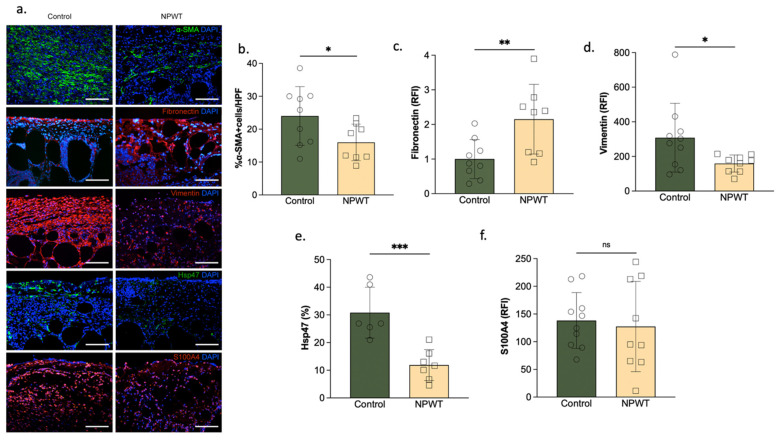
αSMA, Fibronectin, Vimentin, Hsp47 and S100A4 deposition. (**a**) Representative sections. Fibronectin, Vimentin, Hsp47 and S100A4 -stained wound beds of the two groups on day 10. Scale bar = 100 µm. (**b**) αSMA deposition. αSMA deposition was significantly higher in the control group (NPWT:16 ± 5% αSMA + cells vs. control: 24 ± 8%αSMA + cells; *p* = 0.04). (**c**) Fibronectin deposition. Fibronectin deposition significantly differed between the two groups (NPWT:2.2 ± 1 RFI vs. control: 1 ± 0.6; *p* = 0.01). (**d**) Vimentin deposition. Vimentin deposition significantly differed between the two groups (NPWT: 160 ± 50 RFI vs. control: 325 ± 203 RFI; *p* = 0.04). (**e**). Hsp47 deposition. Hsp47 deposition significantly differed between the two groups (NPWT:12 ± 5.6% vs. control: 31 ± 9.2%; *p* = 0.0008). (**f**) S100A4 deposition. No differences were noted in S100A4 deposition between the two groups (NPWT:127 ± 81 RFI vs. control: 138 ± 50 RFI; *p* = 0.73). Number of circles/squares per bar indicates sample size (*n* = 8–9). Each circle/square represents the average of three HPF measurements. Where * equals *p* ≤ 0.05, ** equals *p* ≤ 0.01, and *** equals *p* ≤ 0.001.

**Figure 5 pharmaceutics-14-02125-f005:**
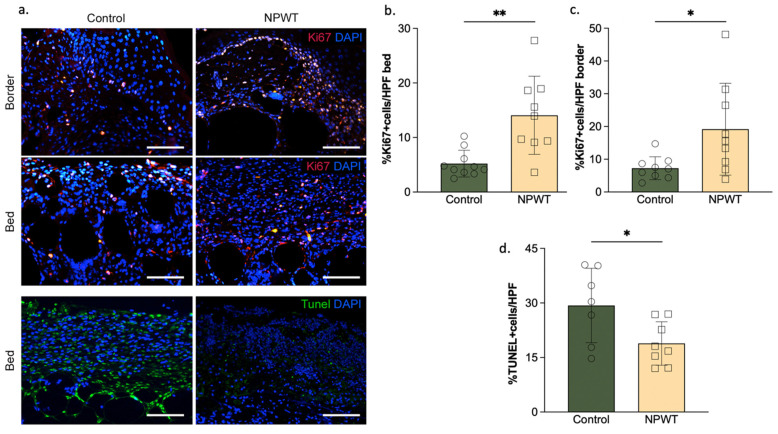
Cell turnover. (**a**) Representative sections. anti-Ki67, Tunel and DAPI stained sections of the wound bed and border of the two groups on day 10. Scale bar = 100 µm. (**b**) Cellular proliferation of the wound bed. Cellular proliferation of the wound bed significantly differed between the two groups (NPWT: 14 ± 7.2% Ki67 + cells/HPF vs. control: 5.2 ± 2.4% Ki67 + cells; *p* = 0.002). (**c**) Cellular proliferation of the wound border. Cellular proliferation of the wound border significantly differed between the two groups (NPWT: 19 ± 14% Ki67 + cells/HPF vs. control:7.3 ± 3.5% Ki67 + cells; *p* = 0.03). (**d**) Cellular apoptosis. Cellular apoptosis significantly differed between the two groups (NPWT: 19 ± 6% TUNEL +/HPF vs. control: 29 ± 10% TUNEL +/HPF; *p* = 0.03). Number of circles/squares per bar indicates sample size (*n* = 8–10). Each circle/square represents the average of three HPF measurements. Where * equals *p* ≤ 0.05 and ** equals *p* ≤ 0.01.

**Figure 6 pharmaceutics-14-02125-f006:**
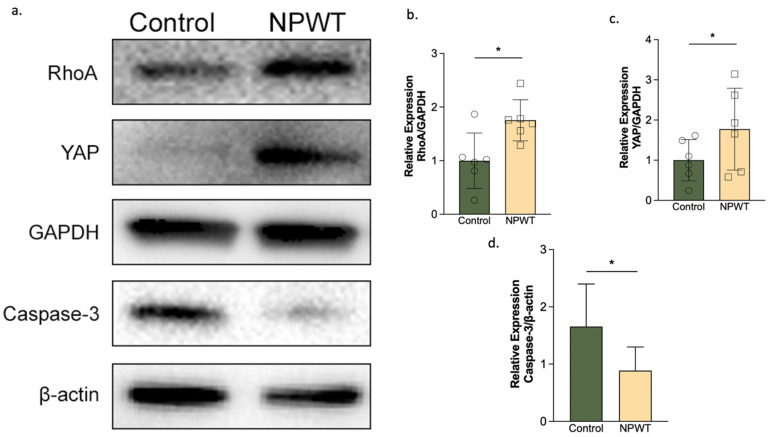
YAP, RhoA and Caspase-3 expression. (**a**) Representative Western blots. YAP, RhoA and Caspase-3 protein expression in the two groups on day 10. (**b**) RhoA relative expression. Densitometry of the ratios of RhoA to GAPDH (*n* = 6; *p* = 0.03) with levels visualized as bar graphs. The NPWT group had significantly higher RhoA levels than the control group. (**c**) YAP relative expression. Densitometry of the ratios of YAP to GAPDH (*n* = 6; *p* = 0.02) with levels indicated as bar graphs. The NPWT group had significantly higher YAP levels than the control group. (**d**) Caspase-3 relative expression. Densitometry of the ratios of Caspase-3 to β-Actin (*n* = 6; *p* = 0.03) with the levels indicated as bar graphs. Caspase-3 levels were significantly higher in the control compared to the NPWT group. Where * equals *p* ≤ 0.05.

**Figure 7 pharmaceutics-14-02125-f007:**
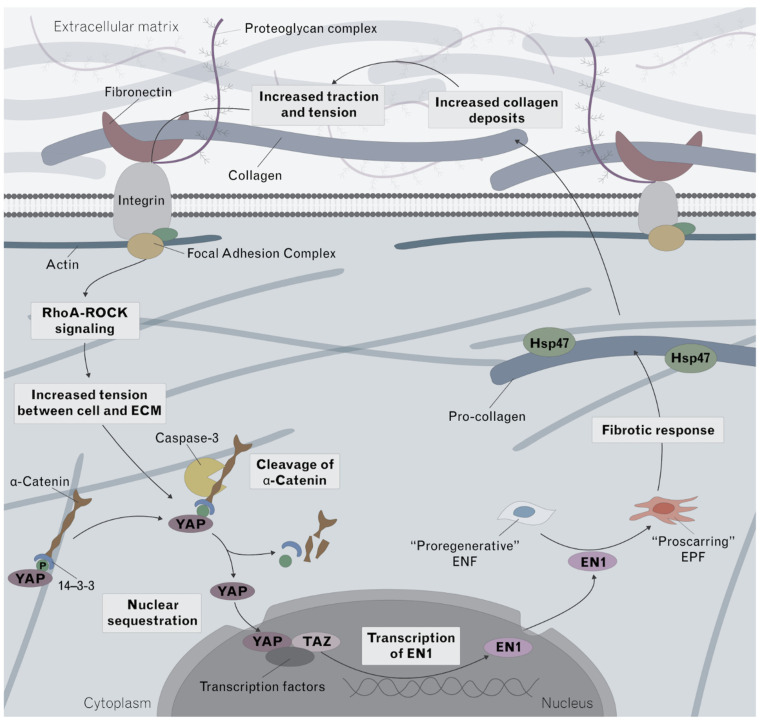
Proposed mechanism of action of NPWT in fibrosis. The normal fibrotic response that occurs during wound healing, and which becomes upregulated in hypertrophic scarring, is summarized in this schematic. Mechanotransduction, in the form of increased tension between the cell and the extracellular matrix, is hypothesized to lead to an upregulation of YAP. Within the cytoplasm, YAP is bound to a-catenin via 14-3-3 which prevents its nuclear sequestration. Fibrosis-promoting signals upregulated caspase-3 which cleaves a-catenin, and allows YAP to translocate into the nucleus. YAP is then able to promote transcription of En1 which induces a switch of the fibroblast phenotype, from pro-regenerative to pro-fibrotic. The fibrotic response results in an increase in the conversion of pro-collagen to collagen. The increased collagen deposition and increase in scarring then further amplifies the loop. In this study, we demonstrate that although NPWT increased the expression of YAP, En1 was decreased. The decrease in Caspase-3 would help explain this de-coupling, as decreased caspase-3 would result in decreased cleavage of a-catenin, decreased nuclear sequestration of YAP and decreased En1 transcription, as identified in our study. A decreased fibrotic response would result in decreased collagen formation. Hsp47 which is required for pro-collagen to be converted to collagen was also identified as decreased in this study, further supporting the decoupling mechanism. YSP, yes-associated protein; EN1, Engrailed-1; RhoA–ROCK, RhoA and Rho-associated protein kinase.

## Data Availability

All efforts were made to provide, throughout the manuscript, details on the specific resources utilized in the study including the animal model, dressings, devices, antibodies, hardware and software. Further details and all data presented in this study are available from the corresponding author upon reasonable request.

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
