# Peer review of "Continuous NPWT Regulates Fibrosis in Murine Diabetic Wound Healing"

_pharmaceutics, 2022, doi:10.3390/pharmaceutics14102125_

Round 1

Reviewer 1 Report

The authors are presenting a review regarding physiological and molecular effect of wound healing when Negative Pressure Wound Therapy (NPWT) is conducted. The article is well presented, however I have some critical points that I feel needs to be corrected before this manuscript can be accepted for publication.

Between different species, processes, molecular pathways, and homologous genes are not always the same. This problem is not often discussed, especially within pharma applications, and I feel that we need to address this matter more. Due to this there might be a drawback when using model organisms, such as mice that is used in the current study, to solve a problem on another organism, human. I would like the authors to address this topic and to which degree human skin models could have been/can be applied instead.

Below are my specific comments and suggestions for improvement.

Throughout the manuscript, please correct the line breaks so that they do not occur in the middle of words. For abbreviations, always write out the full name the initial time that the abbreviation is used, e.g. for BSA at row 147.

Throughout the manuscript, when citing other publications I suggest to write xxxxxyy. Instead of xxxxx.yy

Row 69-70, I suggest to write “Reticular ENFs cultured on soft hydrogels did not show activation of En1, whereas those cultured on high-stiffness plastic showed upregulated En1 expression and transitioned…” instead of “Reticular ENFs cultured on soft hydrogels did not activate En1, whereas those cultured on high-stiffness plastic upregulated En1 expression and transitioned…”.

Row 168, I suggest to write “The wound tissue was homogenized through multiple centrifugations and aspiration of the supernatant” if one aspiration timepoint was used or “The wound tissue was homogenized through multiple centrifugations and aspirations of the supernatant” if several aspiration timepoints were used instead of “The wound tissue was homogenized with multiple centrifugations and aspirations of the supernatant.”  

Row 224-232, Define if these results are statistically significant or not.

Row 230-231, I suggest to write “This suggests that at protein levels, NPWT upregulated the expression of RhoA and YAP, whereas Caspase-3 was downregulated” or similar instead of “This suggests that NPWT up-regulated the expression of RhoA and YAP, and downregulated Caspase-3 at the protein level”.

Author Response

Between different species, processes, molecular pathways, and homologous genes are not always the same. This problem is not often discussed, especially within pharma applications, and I feel that we need to address this matter more. Due to this there might be a drawback when using model organisms, such as mice that is used in the current study, to solve a problem on another organism, human. I would like the authors to address this topic and to which degree human skin models could have been/can be applied instead.

Thank you for this comment. We have added a section to our limitations describing the issues that exist when comparing murine to human wound healing. We also present alternatives, as well as methods we applied to limit the issue as much as possible.

Below are my specific comments and suggestions for improvement.

Throughout the manuscript, please correct the line breaks so that they do not occur in the middle of words. 

Corrected.

For abbreviations, always write out the full name the initial time that the abbreviation is used, e.g. for BSA at row 147.

Corrected.

Throughout the manuscript, when citing other publications I suggest to write xxxxxyy. Instead of xxxxx.yy

Corrected.

Row 69-70, I suggest to write “Reticular ENFs cultured on soft hydrogels did not show activation of En1, whereas those cultured on high-stiffness plastic showed upregulated En1 expression and transitioned…” instead of “Reticular ENFs cultured on soft hydrogels did not activate En1, whereas those cultured on high-stiffness plastic upregulated En1 expression and transitioned…”.

Corrected.

Row 168, I suggest to write “The wound tissue was homogenized through multiple centrifugations and aspiration of the supernatant” if one aspiration timepoint was used or “The wound tissue was homogenized through multiple centrifugations and aspirations of the supernatant” if several aspiration timepoints were used instead of “The wound tissue was homogenized with multiple centrifugations and aspirations of the supernatant.”  

Corrected.

Row 224-232, Define if these results are statistically significant or not.

Defined.

Row 230-231, I suggest to write “This suggests that at protein levels, NPWT upregulated the expression of RhoA and YAP, whereas Caspase-3 was downregulated” or similar instead of “This suggests that NPWT up-regulated the expression of RhoA and YAP, and downregulated Caspase-3 at the protein level”.

Corrected.

Thank you for your review.

Reviewer 2 Report

The manuscript PHARMACEUTICS-1925299 entitled “Continuous NPWT Regulates Fibrosis in Murine Diabetic 2 Wound Healing”by Mengfan Wu et al. investigated the hypothesis that modulation of the wound microenvironment through mechanotransduction forces, such as those applied by NPWT, could be a valid method to minimize pathological scarring. The authors used a murine diabetic excisional model to investigate the effects of NPWT on multiple factors in the pathway of scarring pathology, including YAP, En1, CD26 and α-SMA.

#General Comments

The subject addressed in this work has scientific relevance as it may result in advances in the understanding of scar pathology and contribute to broadening the understanding of the effects of NPWT on multiple factors in the scar pathology pathway, including YAP, En1, CD26 and α-SMA. In general, the work is interesting, the objectives are well defined, and the experimental design is appropriate.

In my opinion, some fundamental aspects were not addressed in the present work.

1)     In recent years, the literature has reported many works dedicated to negative pressure therapy. In this sense, the novelty statement providing information about what is new and innovative in the manuscript about these works would be interesting. It is not clear what is innovative about this work and its impacts in the field. Authors should emphasize this aspect in the text.

2)     Reading some parts of the work reminded me of a cake recipe due to the mechanical way in which the arguments were presented. In the reviewer's opinion, the authors could be more careful in conducting the data discussion. Emphasize the new and important aspects of your study and put your findings in the context of the totality of the relevant evidence. State the limitations of your study and explore the implications of your findings for future research and clinical practice.

3)     The conclusions of the work were not presented. The conclusion is an important part of the article as it provides closure for the reader while reminding them of the content and importance of the article. In some parts of the texts the authors seem to direct the reader to some conclusion, but with little incisiveness in writing.

Author Response

The subject addressed in this work has scientific relevance as it may result in advances in the understanding of scar pathology and contribute to broadening the understanding of the effects of NPWT on multiple factors in the scar pathology pathway, including YAP, En1, CD26 and α-SMA. In general, the work is interesting, the objectives are well defined, and the experimental design is appropriate.

Thank you for your positive feedback.

In my opinion, some fundamental aspects were not addressed in the present work.

1)     In recent years, the literature has reported many works dedicated to negative pressure therapy. In this sense, the novelty statement providing information about what is new and innovative in the manuscript about these works would be interesting. It is not clear what is innovative about this work and its impacts in the field. Authors should emphasize this aspect in the text.

We have revised the manuscript to emphasize the novelty of the study, as well as expanding the innovation section of the manuscript.

2)     Reading some parts of the work reminded me of a cake recipe due to the mechanical way in which the arguments were presented. In the reviewer's opinion, the authors could be more careful in conducting the data discussion. Emphasize the new and important aspects of your study and put your findings in the context of the totality of the relevant evidence. State the limitations of your study and explore the implications of your findings for future research and clinical practice.

Thank you for this feedback, we have edited the discussion section and have added a limitations section. We have attempted throughout the article to highlight what is novel about this study and why it holds clinical promise.

3)     The conclusions of the work were not presented. The conclusion is an important part of the article as it provides closure for the reader while reminding them of the content and importance of the article. In some parts of the texts the authors seem to direct the reader to some conclusion, but with little incisiveness in writing.

We have added a conclusion section as per your feedback.

Round 2

Reviewer 2 Report

The authors satisfactorily answered all the points raised by the reviewer. Therefore, I recommend accepting the work for publication in its present form.